# Prediction of the Hind-Leg Muscles Weight of Yearling Dairy-Beef Steers Using Carcass Weight, Wither Height and Ultrasound Carcass Measurements

**DOI:** 10.3390/ani10040651

**Published:** 2020-04-09

**Authors:** Addisu Hailu Addis, Hugh Thomas Blair, Stephen Todd Morris, Paul Richard Kenyon, Nicola Maria Schreurs

**Affiliations:** Animal Science, School of Agriculture and Environment, Massey University, Palmerston North 4442, New Zealand; H.Blair@massey.ac.nz (H.T.B.); S.T.Morris@massey.ac.nz (S.T.M.); P.R.Kenyon@massey.ac.nz (P.R.K.); N.M.Schreurs@massey.ac.nz (N.M.S.)

**Keywords:** beef, meat yield, model, yearling, young

## Abstract

**Simple Summary:**

Carcass classification and grading systems are typically inadequate for young cattle processed for beef production. Conformation of the hindquarter region of cattle has been used to classify and grade the whole carcass from older beef cattle. This study was initiated with the objective of providing a carcass classification and grading system based on hind-leg muscles weight. Prediction equations for the indirect prediction of saleable meat yield using hind-leg muscles weight from young dairy-origin steers were developed, and could be used for their carcass classification and grading. These equations avoid the need to isolate and track boneless subprimal cuts to establish the saleable meat yield of individual animals.

**Abstract:**

Prediction equations have been widely utilized for carcass classification and grading systems in older beef cattle. However, the equations are mostly relevant for common beef breeds and 18 to 24 month old animals; there are no equations suitable for yearling, dairy-origin cattle. Therefore, this study developed prediction models using 60 dairy-origin, 8 to 12 month old steers to indicate saleable meat yield from hind-legs, which would assist with carcass classification and grading. Fat depth over the rump, rib fat depth, and eye muscle area between the 12th and 13th ribs were measured using ultrasound, and wither height was recorded one week prior to slaughter. The muscles from the hind-leg were retrieved 24 h after slaughter. Prediction equations were modeled for the hind-leg muscles weight using carcass weight, wither height, eye muscle area, rump, and rib fat depths as predictors. Carcass weight explained 61.5% of the variation in hind-leg muscles weight, and eye muscle area explained 39.9% (*p* < 0.05). Their combination in multivariate analysis explained 63.5% of the variation in hind-leg muscles weight. The R^2^ of the prediction in univariate and multivariate analyses was improved when data were analyzed per age group. Additional explanatory traits for yearling steers, including body length, hearth girth, and muscle depth and dimensions measured using video image analysis scanning (VIAscan), could improve the prediction ability of saleable meat yield from yearling dairy beef steers across the slaughter age groups.

## 1. Introduction 

In 2018, global beef production was estimated at 72.1 million tonnes (carcass weight equivalent), representing 22.5% of total meat production [1]. Beef production increasingly utilizes calves originating from the dairy industry [2,3], which is partly due to the expansion of dairy farming producing an accessible supply of calves [4]. In New Zealand, 65% of the annual beef production is sourced from dairy-origin cattle [5], and the dairy industry supplies 35% to 40% of annual calves required for beef finishing [4].

On dairy farms, calves are produced in excess of the dairy herd’s replacement requirements. These calves may go to beef finishing farms or be processed for veal or pet-food [2,5]. In New Zealand, 1.7 million calves from the dairy industry were commercially processed at 4 to 8 days of age in 2018 [5]. There is scope to further increase beef production if more calves from the dairy industry are reared for beef [4,6]. Dairy-origin steers managed in a beef finishing system to 24 to 36 months of age produce beef that has an acceptable eating quality [7,8]. However, due to resource constraints (in particular, grazing land), it is not possible to finish a larger number of calves from the dairy industry for beef to an age of 24 to 36 months. Yearling beef is a potential solution to this issue by accelerating the cycle of beef production. Animals of a similar age are produced in Europe and marketed under different descriptions, such as *Jungrindfleisch* (Austria, Germany), rose veal (Ireland, France), or *carne de ternera* (Spain) [2], and yearling beef production is a common practice in Argentina [9].

Many carcass classification systems for beef indirectly consider saleable meat yield [10,11] based on conformation, muscling, and fat depths in the hindquarter region, e.g., the EUROP carcass classification system or the fat depth by conformation matrix used in New Zealand [5,12,13,14]. However, the carcass classification schemes for yearling cattle are generally undefined [2], and those used for older cattle are unlikely to be applicable due to a less developed conformation and fat deposition in yearling cattle [15]. Therefore, there is a need to provide carcass classification and grading systems for creating a fair payment system.

To measure the saleable meat yield, boneless primal and subprimal cuts are obtained; these can be costly and require substantial labor [12,16,17]. For countries that do not use video image analysis (VIA) scanning to assess saleable meat yield [18], prediction equations are a practical tool. The use of prediction equations avoids the need to assess saleable meat yield directly (thereby avoiding any effect on the quality of the meat), and could be used to assess potential meat yield on live animals if ultrasound measures of carcass composition are used as the predictors [12,17,19,20].

Carcass weight, eye muscle area, rump, and rib fat depths have been used as predictors of beef saleable meat yield [14,17,21,22] but lack applicability across age, breed, and gender variables [23]. Predictive models for saleable meat yield have not yet been developed for yearling steers. The boneless muscle weight from the hindquarter region has been used as an indicator of saleable meat yield for the whole carcass [12,13,14,24]. Hence, this study was initiated to identify the most pertinent explanatory variables for the weight of boneless cuts from the hind-legs of 8, 10, and 12 month old steers, and to develop predictive models which could be employed for both whole carcass classification and grading, thereby allowing the assignment of a carcass value. It was hypothesized that a combination of carcass weight, wither height, ultrasound eye muscle area, rump, and rib fat depths would be suitable predictors of hind-leg muscles yield, and would elucidate characteristics of importance for use in a carcass classification system for yearling cattle.

## 2. Materials and Methods

This study utilized data from the study of Pike et al. [25] for the development of prediction equations. The animals and data collection for this study are briefly described here.

### 2.1. Animals

There were 60, 3 month old Hereford, Jersey, Holstein Friesian crossbred calves (103 ± 11 kg live weight) that were obtained from a commercial rearer, and randomly assigned into a slaughter age group of 8, 10, or 12 months of age. The steers were managed together as one group at Massey University’s Keeble farm. They grazed herb mixes containing plantain (*Plantago lanceolata*), chicory (*Cichorium intybus*), white clover (*Trifolium repens*), and red clover (*Trifolium pratense*), with 0.5 kg meal/head for 2 months after arrival on the farm. This was followed by leafy turnip (Ceres Hunter, Forage Brassica, Agricom NZ) for a month, with the remaining time up to slaughter on a perennial ryegrass-based pasture. To ensure intake was not restricted, cattle had *ad libitum* forage allowance, and the forage mass did not go below 1200 kg dry matter/hectare (kg DM/ha) at any time during grazing [4,26], which allowed for a live weight gain of approximately 0.9 kg/head/day. The study was completed with the approval of the Massey University Animal Ethics Committee (MUAEC 17/73).

### 2.2. Data Collection

Ultrasound measurements of the carcass characteristics were collected in the week before slaughter at a site on the animal between the 12th and 13th ribs for rib fat depth (RF, mm) and eye muscle area (EMA, cm^2^), and on the rump for fat depth (P8, mm) [19,21]. Wither height was measured at the time of ultrasound measurements. Hot carcass weight was obtained after commercial dressing procedures. Carcasses were chilled for 24 h and knuckle, topside, and silverside cuts were retrieved. The total hind-leg muscles weight was calculated by summing up the weights of the three cuts, i.e., topside, knuckle, and silverside, representing the major muscles surrounding the femur and the associated fat. The knuckle contains the quadriceps femoris muscle; the silverside (also called the outside round) includes the biceps femoris and semitendinosus muscles; and the topside contains the semimembranosus, adductor, and pectineus muscles [24].

### 2.3. Data Analysis

All analyses were conducted using the R software, version 3.6.0 [27]. Descriptive statistics and analysis of variance (ANOVA) for the slaughter age groups were undertaken. The association between the dependent variable, hind-leg muscles weight (LM), and the independent variables within a slaughter group, and for the data from three slaughter groups as a single group, were tested using Pearson correlations. The predictors of hind-leg muscles weight included: carcass weight (CW), wither height (WH), eye muscle area (EMA), rump fat depth (P8), and rib fat depth (RF), and were fitted in two models: linear mixed models (LMM) and generalized least squares (GLS) using the nlme extension [28]. The maximum likelihood estimation method was used in both models and the corresponding models were compared. The best-fit model was that with a lower Akaike’s Information Criterion (AIC). Residual errors in the final models were evaluated for influential outlier effects using Cook’s distances [29]; normality using Q-Qplots [30]; and multicollinearity using a variance inflation factor (VIF) [31]. The multivariate analysis was carried out using forward selection regression [19]. The univariate and multivariate models were validated using goodness of fit (R^2^ value) and prediction accuracy metrics, including root mean square error (RMSE) [32].

## 3. Results

### 3.1. Carcass Yield and Compositions

The 12 month old steers produced a heavier carcass and had greater wither height, eye muscle area, rump, and rib fat depths (*p* < 0.05, Table 1). Hind-leg muscles weight was lower in the 8 month old steers (*p* < 0.05, Table 1).

### 3.2. Relationship between Independent and Dependent Variables

Carcass weight had a positive correlation with hind-leg muscles weight across the slaughter groups and within groups (correlation coefficient of 0.79 to 0.91, Table 2). Hind-leg muscles weight was correlated with all independent variables in the 10 month old steers and in the combined data. However, the hind-leg muscles weight did not correlate with fat depths in the 8 and 12 month old steers (Table 2).

### 3.3. Predictive Equations for Hind-Leg Muscles Weight

For the data across all slaughter ages, carcass weight, wither height and eye muscle area were significant predictors of hind-leg muscles weight in yearling steers (*p* < 0.05; Table 3). Carcass weight explained 61.5% of the variability in hind-leg muscles weight, whereas eye muscle area explained only 39.9% of this variation. Their respective prediction accuracies were 0.91 and 1.14 (Table 3). The multivariate analysis indicated that 65.7% of the variation in hind-leg muscles weight was explained by using carcass weight, wither height, and eye muscle area.

When using data from each slaughter age treatment separately, the goodness of fitness as well as the prediction accuracies were improved, compared to using data from all age groups. Carcass weight explained from 68.7% to 82.8% of variation in hind-leg muscles weight within the slaughter age groups (Table 3). The corresponding prediction accuracies were 0.46 and 0.44 (Table 3). Eye muscle area was a significant univariate predictor in 8 and 10 month old steers whereas wither height was significant in the 12 month old steers (Table 3). In total, 71.3%, 91.9%, and 72.4% of the variation in hind-leg muscles weights from 8, 10, and 12 month old steers, respectively, were explained using multivariate analysis (Table 3).

## 4. Discussion

Hind-leg muscles weight represents the major muscle from beef carcasses and is typically employed as an indicator of total saleable meat yield. This study was initiated with the main objective of developing predictive equations for hind-leg muscles weight in yearling steers to assess saleable meat yield, and to identify those variables that could be implemented in a classification system. A strong positive correlation between carcass weight and saleable meat yield was translated into prediction equations in older cattle [10,16,22,33], and this was also evident for the yearling cattle in the current study.

The 12 month old steers were heavier and produced higher carcass weight than younger steers. They also had a higher wither height, although it was lower than the 121.8 cm reported in a mature cow [34]. Bone is the earliest developing tissue, followed by muscle and then fat [15,35]. Muscle mass increases with carcass weight at a diminishing rate and then plateaus at the fattening stage [15,35]. The 10 and 12 month old steers produced a heavier hind-leg muscles weight than did the 8 month old steers, although it was lower than the 13.3 kg reported from different breeds of steers up to 3 years of age [24]. The eye muscle area of yearling steers was 65.5% of the size reported in Hereford-sired dairy-Angus crossbred steers at 22 to 25 months of age (59.2 cm^2^ to 75.3 cm^2^) [8]. Tarouco et al. [14] reported 70.8 cm^2^ of eye muscle area in Nellore steers at 24 to 30 months of age. Bergen et al. [19] and Lee et al. [11] reported an eye muscle area of 96 cm^2^ in one-year old bulls and Hanwoo steers at 32 months of age. The lower value of muscle weights in the yearling steers in this study is due to the younger age and lighter weight at slaughter.

Regardless of breed, gender, and nutrition status of an animal, fat growth is faster as animals approach maturity [15,35]. The 12 month old steers had thicker rump and rib fat depths than the young steers, but were half that reported in 22 to 25 month old Hereford-sired dairy-Angus crossbred steers [8]. Bergen et al. [19] reported 6.0 mm and 5.1 mm rump and rib fat depths, respectively, in one-year old bulls. In Nellore steers at the age of 24 to 30 months, a 9.2 mm rump fat depth and 6.4 mm rib fat depth were reported [14]. The lower fat depths in this study were likely due to the growth stage of the steers and the predominately forage diet. Animals deposit more fat when fed a diet composed of concentrates, rather than pasture or forages [15].

For the 8 to 12 month old steers in the current study, carcass weight explained the largest proportion of the variation in hind-leg muscles weight. The coefficient of determination (R^2^ value) and prediction accuracy (RMSE) of carcass weight within each slaughter treatment group were increased up to 25.7% and 55.6% of its respective value in the combined data. Similar to our study, Chen et al. [22] reported that carcass weight could explain 63% to 90% of the variation in the weight of trimmed top-grade cuts from native and crossbred Chinese Yellow steers at age of 18 to 52 months. Epley et al. [33], Berry et al. [36], and Lee et al. [11] reported 83.4% to 86.0% for the coefficient of determination of carcass weight in predicting beef prime cuts. In agreement with the current study, several studies have identified carcass weight as being the strongest predictor of beef meat yields [11,16,22,33,36].

Eye muscle area and fat depths assessed by ultrasound were more accurate (higher correlation coefficient) than the carcass measured equivalents in predicting saleable meat yield [14,19,20]. Amongst the ultrasound measurements, only eye muscle area was a significant predictor of hind-leg muscles weight from the yearling steers, although it controlled the least variation (39.9%). Similarly, Greiner et al. [37] reported that the ultrasound eye muscle area controlled 37% of the variation in beef saleable meat yield from 1 to 2-year old steers. Similarly, in a study with different age classes of Angus and Angus-crossbred bulls and steers, 41% of the variation in beef saleable meat yield was explained by the ultrasound eye muscle area [20]. Neither rump nor rib fat depth was a significant predictor of hind-leg muscles weight in yearling steers; they were also not significant predictors of weight of saleable meat from the hind-legs in Nellore steers at 24 to 30 months old [14]. Fat measurements have previously been correlated to the percentage of saleable meat yield rather than the weight [14,22].

The coefficient of determination and accuracy of prediction equations were improved with multivariate analysis. The prediction efficiency and accuracy between models using carcass weight and wither height or carcass weight and eye muscle area, across the age groups, were not different. The prediction abilities of multivariate analysis in terms of the R^2^ value and accuracy were improved when using data within each of the slaughter age groups. Epley et al. [33] from mixed carcasses, and Lee et al. [11] in 32 month old Hanwoo steers, reported R^2^ values of 88.0% and 85.9% using carcass weight and eye muscle area in predicting valuable beef cuts, respectively. According to Brungardt and Bray [38], 82.0% of the variation in the saleable meat yield retrieved from four wholesale cuts of steers was explained using the percent of kidney fat, left side carcass weight, eye muscle area, and percent of trimmed round yield. With different breeds of steers considered, 94% of the variation in beef saleable meat yield was explained using eye muscle area, side carcass weight, and trimmed round-cut as predictors [16].

The goodness of fitness and accuracy of the models using the combined data from all the slaughter age groups for hind-leg muscles weight were lower than the corresponding models within the slaughter age group. Therefore, it is recommended to use the within-slaughter age prediction equations; if an accurate record of age is known when slaughtered at less than 12 months of age, it would be preferable to validate these equations with larger data sets before application. However, if the carcasses are from steers of approximately one year of age, but the exact age is not known, the prediction equations developed utilizing the data across the slaughter ages could be used. The prediction equations developed in this study could be used for beef classification systems which utilize cattle that originate from dairy farms, are finished in pasture-based production systems, and are processed at an age of approximately 8 to 12 months [2].

## 5. Conclusions

In conclusion, carcass weight can be used to assess boneless muscle weight of hind-legs from yearling dairy-origin steers, and may be useful for developing a classification scheme to assign a value for their carcasses. Its prediction efficiency and accuracy increased when combined with wither height and eye muscle area measured by ultrasound. To improve the overall prediction ability and accuracy in classifying meat yield across the slaughter age groups, additional traits are likely to be required. From the live animal, variables such as body length, hip width, and hearth girth could be considered, although these are less practicable to obtain on a large number of cattle. On the carcass, muscle depth and dimensions measured using VIAscan might be useful; however, VIAscan did not improve prediction equations in lamb [39] and beef [40].

## Figures and Tables

**Table 1 animals-10-00651-t001:** Descriptive statistics for live weight, carcass weight, wither height, hind-leg muscles weight, eye muscle area, rump and rib fat depths, measured on Hereford–Kiwi (Jersey, Holstein Friesian) crossed steers at 8 (*n* = 20), 10 (*n* = 20) and 12 (*n* = 20) months of age.

Attributes	Slaughter Age (Months)											
	8				10				12			
	Mean	sd.	Range		Mean	sd.	Range		Mean	sd.	Range	
Live weight (kg)	252.2 ^c^	25.0	214.0	300.0	302.8 ^b^	17.2	272.0	335.0	347.6 ^a^	22.0	303.0	392.0
Carcass weight (kg)	119.0 ^c^	12.3	97.7	141.1	145.5 ^b^	13.0	124.7	179.1	173.9 ^a^	11.0	148.8	193.9
Wither height (cm)	108.1 ^c^	3.0	103.0	114.0	116.6 ^b^	4.1	108.0	124.0	119.7 ^a^	3.1	115.0	126.0
Hind-leg muscles weight (kg)	9.0 ^b^	0.8	7.6	10.7	11.3 ^a^	1.1	9.2	13.5	11.2 ^a^	1.2	8.2	13.6
Eye muscle area (cm^2^)	38.2 ^c^	3.1	34.0	45.0	41.8 ^b^	3.7	35.0	49.0	51.2 ^a^	4.1	46.0	62.0
Rump fat depth (mm)	1.8 ^b^	0.8	1.0	4.0	2.1 ^b^	0.6	1.0	3.0	2.8 ^a^	0.9	2.0	5.0
Rib fat depth (mm)	1.1 ^c^	0.3	1.0	2.0	1.6 ^b^	0.5	1.0	2.0	2.1 ^a^	0.3	2.0	3.0

^a,b,c^ Superscript within row trait indicates means are significantly different at *p* < 0.05; sd.: standard deviation

**Table 2 animals-10-00651-t002:** Pearson correlation coefficient of hind-leg muscles weight (LM, kg) with carcass weight (CW, kg), wither height (WH, cm), eye muscle area (EMA, cm^2^), rump (P8, mm) and rib fat depths (RF, mm) in 8 (*n* = 20), 10 (*n* = 20), and 12 (*n* = 20) month old steers and for data of all ages

Dependent Variable, LM	Independent Variables				
	CW	WH	EMA	P8	RF
Slaughter age (months)					
8	0.82 ***	0.01	0.75 ***	0.17	0.26
10	0.91 ***	0.55 *	0.73 ***	0.54 *	0.47 *
12	0.85 ***	0.46 *	0.34	−0.40	−0.23
All ages ^†^	0.79 ***	0.71 ***	0.63 ***	0.26 *	0.55 ***

*** *p* < 0.000; * *p* < 0.05; ^†^ Combined data from 8, 10, and 12 month old steers.

**Table 3 animals-10-00651-t003:** Linear models for hind-leg muscles weight using carcass weight (CW, kg), wither height (WH, cm), eye muscle area (EMA, cm^2^), rump fat depth (P8, mm), and rib fat depth (RF, mm) for the combined steer data and within slaughter age group.

Slaughter Age Group (Months)	Intercept	Partial Regression Coefficients					R^2^	RMSE
		CW	WH	EMA	P8	RF		
All ages ^†^								
	3.83	0.05	__	__	__	__	61.5	0.91
	−9.88	__	0.18	__	__	__	51.2	1.03
	4.37	__	__	0.14	__	__	39.9	1.14
	−1.52 ^‡^	0.03	0.06 ^§^	__	__	__	64.2	0.88
	4.43	0.06	__	−0.06 ^§^	__	__	63.5	0.89
	−0.79 ^‡^	0.05	0.06 ^§^	−0.06 ^§^	__	__	65.7	0.86
8								
	2.26	0.06	__	__	__	__	68.7	0.46
	1.16	__	__	0.21	__	__	56.0	0.55
	1.19 ^‡^	0.04	__	−0.07 ^§^	__	__	71.3	0.44
10								
	−2.11 ^‡^	0.09	__	__	__	__	82.8	0.44
	2.54 ^‡^	__	__	0.21	__	__	52.7	0.73
	−9.23	0.08	0.07	__	__	__	88.8	0.35
	−10.30	0.09	0.08	0.01 ^§^	0.18 ^§^	−0.47 ^§^	91.9	0.30
12								
	−4.80 ^‡^	0.09	__	__	__	__	71.5	0.62
	−10.4 ^‡^	__	0.16	__	__	__	21.4	1.03
	−8.91 ^‡^	0.09	0.04	__	__	__	72.4	0.61

^†^ Combined data from 8, 10, and 12 month old steers at slaughter; ^‡^ The intercepts were not significantly different from zero at *p* > 0.05; ^§^ independent variables were not significant in the multivariate analyses at *p* > 0.05; R^2^, coefficient of determination; RMSE, root mean square error.

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
