# Peer review of "Prediction of the Hind-Leg Muscles Weight of Yearling Dairy-Beef Steers Using Carcass Weight, Wither Height and Ultrasound Carcass Measurements"

_animals, 2020, doi:10.3390/ani10040651_

Round 1

Reviewer 1 Report

The manuscript is presented as a well written scientific article. English language is appropriate and understandable.

In title, I recommend adding a “s” after muscle, since it refers to several muscles.

The paper seems to be original in general terms.  It has a strong justification based on the need to provide carcass classification and grading systems for young and dairy-breeds steers to create a fair payment system.

    The prediction of carcass composition is very important for value-based marketing, and the improvement of prediction accuracy and precision can be achieved through the analyses of independent variables using a prediction equation with enough dataset. In this study, the number of animals used within each age group is small to represent each subgroup for a predicted equation.  Between lines 224- 227, the authors said: “Therefore, it is recommended to use the within slaughter age prediction equations, if an accurate record of age is known when slaughtered at less than months of age”. The authors should explain how they recommend a prediction equation developed with 20 observations. They could indicate that their results are preliminary and that need validation with a higher number of observations.

Prediction equations were obtained only for hind-leg muscle expressed in kgs. The authors did not show predicted equations of the hind-leg muscle expressed in percentage (%). It is likely that the use of predicted equation (expressed in kgs) will be restricted to a certain carcass weight range of the population (as they present in discussion section). This would limit its recommendation as a method of carcasses classification system. In discussion, also it is necessary to present the reason to present a prediction equation for for hind-leg muscle expressed in kgs.

Reviewer 2 Report

Review

Animals -762294

Title: Prediction of the hind-leg muscle weight of yearling dairy-beef steers using carcass weight, wither height and ultrasound carcass measurements

General comments:

The study aims to develop predictive models combining measurable variables such as carcass weight, wither height, eye muscle area, rump, and rib fat, among others, for use in a carcass classification system for cattle.

In the abstract, it is stated that “ Carcass weight explained 61.5% of the variation in hind-leg muscle weight and eye muscle area explained 39.9% (P<0.05)” which comes from the results. It is slightly over the Random guess. Could you please enhance the reason we should accept your proposition for adopting your solution?

Please check:

Lines 87-88: please use the Latin words in italic font.

Line 156, in Table 3: please check the proper citation of cm2.

Line 160: please check R2.

Line 174: cm2.

Lines 227-229: please state here the limitations of the findings. Please check the comment above.

Reviewer 3 Report

Simple summary:

L13 providing a carcass classification system

L15-16 this sentence overlaps mostly with the former sentence; I suggest shortening it to 'These prediction models could be used for carcass classification and grading of young Dairy-origin steer beef'

Abstract: 

L20 systems

L23 from 

L31 ….were improved: maybe this sentence could be clarified by changing it into 'The R2 of the prediction in univariate and multivariate analyses were improved when data were analysed per age group'

L33 any idea as to what additional traits would improve prediction?

Materials and methods

L84 103 +- 1 kg, is that variation correct? It seems very small. 

L110 For more clarity in this sentences, I would suggest small changes: 'The predictors of hind leg muscle weight included carcass weight (CW),...., and rib fat depth (RF) and were fitted in two models: linear mixed models (LMM) and generalized least squares (GLS), using the nlme extension'  

Results

L122-124 It seems too logical that older steers are bigger and heavier. Why report this as a result? Weights are already in Table 1. Maybe just refer to the Table. 

L147 for clarity, I suggest: When using data from each slaughter-age treatment separately, ...

L148 ...were improved compared to using data...

Discussion

L 167 What is meant by 'a higher carcass'?

L171 I suggest placing the comma after 'steers': 8-month-old steers, although it was lower… 

L177 I suggest putting this conclusion stronger, as it is obvious: ….in this study is due to the younger age... 

L235 Any suggestions about what extra trait would be useful to add? 

Round 2

Reviewer 1 Report

I am very pleased with the Author´s responses.

A few minor aspects to attend:

Lines 199, 219 and 220: “et al”. is in italic, it has to be changed to normal.

Table 1: Change “P<0.05” to “p<0.05”. As a comment, the numbers in the first row are very close to each other, difficult to read.

Table 3: Change "12 months old to “12-months-old

Lines 105-107: muscles´ names are not in italic.

Author Response

We would like to appreciate the reviewer for his/her precious time and comments which have improved the manuscript a lot.

Below are Reviewer`s comments and Authors` responses.

Reviewer: Lines 199, 219 and 220: “et al”. is in italic, it has to be changed to normal.

Authors: L199, 219, 220 correction accepted

Table 1: Change “P<0.05” to “p<0.05”. As a comment, the numbers in the first row are very close to each other, difficult to read.

Authors: L131 correction accepted and Table 1 has been reformatted to landscape to enhance the readability of it`s content

Table 3: Change "12 months old to “12-months-old

Authors: L161 correction accepted based on reviewer`s comment

Lines 105-107: muscles´ names are not in italic.

Authors: L105-107 muscle names were italicized

Thank you for your valuable time and comments

Authors